# Taurine Supplementation Alleviates Blood Pressure via Gut–Brain Communication in Spontaneously Hypertensive Rats

**DOI:** 10.3390/biomedicines12122711

**Published:** 2024-11-27

**Authors:** Qing Su, Xiong-Feng Pan, Hong-Bao Li, Ling-Xiao Xiong, Juan Bai, Xiao-Min Wang, Xiao-Ying Qu, Ning-Rui Zhang, Guo-Quan Zou, Yang Shen, Lu Li, Li-Li Huang, Huan Zhang, Meng-Lu Xu

**Affiliations:** 1Department of Physiology and Pathophysiology, School of Basic Medical Sciences, Xi’an Jiaotong University, Xi’an 710061, China; qingsuxjd2018@xjtu.edu.cn (Q.S.); hongbaoli1985@163.com (H.-B.L.); w312683766@163.com (X.-M.W.); 2204521752@stu.xjtu.edu.cn (L.-L.H.); 2Pediatrics Research Institute of Hunan Province, Hunan Children’s Hospital, Changsha 410007, China; pxfcsu@163.com; 3Department of Urology, The First Affiliated Hospital of Xi’an Jiaotong University, Xi’an 710061, China; xlx3721@stu.xjtu.edu.cn; 4Department of Anesthesiology & Center for Brain Science, The First Affiliated Hospital of Xi’an Jiaotong University, Xi’an 710061, China; baijuan321123@163.com; 5Department of Clinical Medical, Xi’an Jiaotong University, Xi’an 710061, China; xiaoyinghl@stu.xjtu.edu.cn (X.-Y.Q.); 2206124265@stu.xjtu.edu.cn (N.-R.Z.); 2806561390@stu.xjtu.edu.cn (G.-Q.Z.); 2226114789@stu.xjtu.edu.cn (Y.S.); 6Department of Nephrology, The First Affiliated Hospital of Xi’an Medical University, Xi’an 710077, China; liluwenqi@163.com; 7Department of Cardiovascular Medicine, The Affiliated Hospital of Northwest University & Xi’an No.3 Hospital, Xi’an 710018, China; doczhanghuan@163.com

**Keywords:** taurine, paraventricular nucleus, gut–brain, tryptophan metabolism

## Abstract

Objects: Taurine exhibits protective effects in the context of cardiovascular pathophysiology. A range of evidence suggests that hypertension activates inflammatory responses and oxidative stress in the paraventricular nucleus (PVN), elevating the arterial tone and sympathetic activity, while it induces gut–brain axis dysfunction in the context of hypertension. However, the mechanism underlying taurine’s anti-hypertensive effects via the gut–brain axis remains unclear. Method: Male spontaneously hypertensive rats (SHRs) were administered 3% taurine in their drinking water for eight weeks, with their arterial pressure measured weekly. Molecular techniques were employed to investigate taurine’s effects on the hypertensive gut and PVN. Additionally, 16S rRNA gene sequencing was used to analyze the gut microbiota composition, and untargeted metabolomics was applied to assess the fecal metabolites following taurine supplementation. Results: Taurine supplementation not only reduced the blood pressure, sympathetic activity, and inflammatory and oxidative stress in the PVN but also improved the cardiac pathology and microbiota composition while alleviating gut inflammation in hypertensive rats. The untargeted metabolite analysis indicated that the primary effect of the taurine intervention in SHRs was exerted on tryptophan metabolism. The levels of serum metabolites such as kynurenine, L-tryptophan, serotonin (5-HT), and 5-hydroxyindole-3-acetic acid (5-HIAA) were altered in hypertensive rats following taurine treatment. Conclusions: Taurine supplementation restored the microbiota balance, strengthened the mucosal barrier, reduced intestinal inflammation, and stimulated tryptophan metabolism. The metabolites derived from the gut microbiota likely crossed the brain barrier and reached the paraventricular nucleus, thereby reducing the inflammatory responses and oxidative stress in the PVN via gut–brain communication, leading to decreased sympathetic nerve activity and blood pressure in the studied hypertensive rats.

## 1. Introduction

Hypertension is a cardiovascular syndrome with the trait of persistently elevated arterial blood pressure. As the pathogenesis of hypertension is complicated and multifactorial, decreases in blood pressure and sympathetic nerve excitation are vital [1,2,3]. Evidence obtained from numerous investigations has proven that dietary taurine supplementation has positive effects on hypertension through the promotion of inhibitory neurotransmitter production, as well as increasing anti-inflammatory and antioxidant levels, protecting cardiomyocytes, and supporting the central nervous system, as demonstrated in several rat models of hypertension [4,5,6]. In addition, it is well established that taurine can protect the myocardial, blood vessel, and blood circulation systems. For example, taurine transporter (Taut) knockout mice with low-taurine diets exhibited an increased incidence of hypertension and cardiac diseases [7,8,9]. The potential application of taurine in hypertension therapy has attracted much more attention. Prior to clinical application, it is necessary to explore the effects of taurine on the intestinal flora and metabolites, as well as the specific mechanisms by which the related metabolites are involved in hypertension.

Recently, new evidence has emerged that demonstrates that the gut microbiota and its metabolites are deeply involved in cardiovascular diseases [10,11,12,13]. The composition of the gut microbiota was altered in cardiovascular diseases such as hypertension, heart failure, arterial disease, and arrhythmia [14,15]. For instance, the abundance of the intestinal flora in several animal models, such as spontaneous hypertensive rats (SHR), rats with salt-induced hypertension, and rats with chronic kidney disease (CKD) and obesity, exhibited significant differences compared with control animals [16,17,18]. The alpha diversity of the gut microbiota, which is considered to be a negative factor regarding blood pressure, is lower in hypertensive rats [19]. Abnormal distribution in the gut microbiota induces imparity in the capacity for metabolite release, including neurotransmitters, norepinephrine, and several vitamins and short-chain fatty acids, which regulate arterial pressure and influence cardiovascular responses [12,20,21,22,23,24]. Furthermore, the gut microbiome and metabolites likely influence blood pressure and sympathetic activity through gut–brain communication; this may include central cross-talk with the microbiota, neuroendocrine–HPA interactions, microbiota and immune responses, neurotransmitters from the gut microbiota, and intestinal mucosa–brain barrier communication [25,26,27,28,29,30,31,32]. The enteric nervous system is located in the submucosal meridian cluster; it directly controls intestinal movement and is responsible for the neural connections between microorganisms and hosts [33]. Moreover, gut microbes exchange sensory information with their hosts by producing a large number of metabolites in the intestinal lumen. Some of them cross the blood–brain barrier and enter the brain [34]. In this communication network, the brain influences gut movement, perception, and secretion functions, but visceral signals from the gut also influence brain function [35].

Our previous studies indicated that hypertension augmented the levels of inflammatory cytokines and reactive oxygen species in the paraventricular nucleus of the hypothalamus (PVN), which is a vital region for blood pressure regulation [36,37]. The bilateral paraventricular nucleus (PVN) inhibition of pro-inflammatory cytokines led to a sharp decrease in heart fibrosis, arterial blood pressure, oxidative stress, and sympathetic activity in a hypertensive animal model [38]. Moreover, the microinjection of AAV-Ec-SOD (Ec-SOD overexpression) in rats with high-salt-induced hypertension eliminates the excessive reactive oxygen species (ROS) in the PVN and lowers the arterial blood pressure, likely through suppressing inflammatory responses and inhibiting the sympathetic outflow in the PVN [39]. Thus, the inhibition of inflammation and oxidative stress is a potential strategy to decrease blood pressure. However, there is little research exploring whether taurine supplementation can suppress the hypertensive response via gut–brain communication. This would be of great significance regarding the clinical application of taurine for hypertension therapy. Therefore, the purpose of this research is to explore the mechanism by which taurine supplementation reduces blood pressure in spontaneously hypertensive rats via the gut–brain axis, which could provide new theoretical support for the clinical application of taurine.

## 2. Material and Methods

### 2.1. Animals

Male SHR and Wistar Kyoto (WKY) rats weighing 150–200 g were obtained from the experimental animal center of Xi’an Jiaotong University and Beijing Vital River Laboratory Animal Technology Co., Ltd. (Beijing, China). Animals were provided free water with a temperature-controlled (23 ± 2 °C) and light/dark cycle environment. All experimental procedures adhered to the Guide for the Care and Use of Laboratory Animals (National Institutes of Health Publication No. 85-23, revised 1996) and were approved by the Animal Care and Use Committee of Xi’an Jiaotong University (No. 2018-404).

### 2.2. General Experimental Protocol

Male SHR rats and WKY weighing approximately 250 g in each cage received 3% taurine (3 g taurine dissolved in 97 mL drinking water, T0625, Sigma-Aldrich, St. Louis, MO, USA) for eight weeks [40,41]. Meanwhile, each water would be refreshed daily. The groups were divided as follows: the WKY, SHR, and SHR+Taurine. The systolic pressure (SBP) was measured weekly using a tail-cuff system. When this study was terminated, the rats were anesthetized with a ketamine (80 mg/kg) and xylazine (10 mg/kg) mixture (i.p.) while using carotid artery intubation to measure the mean arterial blood pressure (MAP). Then, fresh tissue, feces, and plasma samples were collected and stored at −80 °C or in 4% paraformaldehyde for the subsequent experiments.

### 2.3. Blood Pressure Measurement

The arterial pressure was recorded by a non-invasive system. It was monitored weekly throughout the eight-week period. Before taurine supplementation, all animals were acclimated to this instrument for a week. An awake rat was placed in a holding device (ambient temperature of 30 °C). The rat’s tail, fitted with a cuff and sensor, was allowed to adapt to the inflation and deflation process for 10 min. When the animals became peaceful and the recording system showed jagged waveforms, the systolic arterial pressure began to record 20 times within a 40 min window between 8:00 a.m. and 11:00 a.m. on each measurement day. The MAP was determined via the intubation of the left common carotid artery for five minutes [39,42,43].

### 2.4. Histology Analysis

Heart and gut tissues were dissected and immersed in 4% paraformaldehyde for 48 h to undergo rapid fixation. Subsequently, tissues were embedded in paraffin after ethanol concentration dehydration and xylene transparent, then sliced into pieces 5 μm thick. All of those slices were soaked in a heat-induced epitope repair (HIER) solution and boiled in the microwave for 10 min. After dewaxing and PBS washing, all of those slices were stained with Masson’s trichrome and hematoxylin–eosin (H&E) according to the specification. For the Masson’s staining, the slices were stained with hematoxylin and ponceau. The collagen fibers were stained blue, while the cytoplasm was stained red. All images were employed for visualization under a microscope (Eclipse 80i, Nikon, Minato, Tokyo) [17,44].

### 2.5. Immunofluorescence and Immunohistochemistry Staining

Paraffin-embedded brain and colon samples were sectioned into several 5 μm transverse sections. The transverse brain sections ranged from bregma −0.92 to −2.12 mm, obtained using a paraffin slicer (RM2016, Leica, Wetzlar, Germany). After the above dewaxing, the brain and colon sections were dropped 3% H_2_O_2_ for 10 min to remove endogenous peroxidase. Then, the heterogenetic antigen was blocked by 5% goat serum dissolved into TBST at room temperature for 1 h. After throwing liquid, the primary antibody in a TBST was incubated at 4 °C overnight. The primary antibodies used in this study are as follows: ZO-1 (1:200, sc-33725, Santa Cruz Biotechnology, Santa Cruz, CA, USA), Occludin (1:200, sc-133256, Santa Cruz), IL-1β (1:100, sc-32294, Santa Cruz), p47^phox^ (1:200, sc-17844, Santa Cruz), and gp91^phox^ (1:200, sc-130543, Santa Cruz). After being washed three times, the secondary antibodies (Alexa Fluor-conjugated donkey anti-mouse or anti-rabbit IgG (1:200; Molecular Probes, Oregon Eugene, OR, USA) were incubated for 1 h in the dark box. Then, sections were added to the DAPI (Invitrogen, Waltham, MA, USA) for 30 min. In order to protect the fluorescence fading, an anti-fade solution was also added. Finally, the immunofluorescence-stained sections were dropped the glycerin and covered with slip. The immunofluorescence images were observed by a fluorescence microscope (Nikon eclipse, 80i, Tokyo, Japan).

For the immunohistochemistry, after primary antibody staining, the sections were incubated with a secondary antibody, anti-rabbit IgG (HRP) (1:200, ab7090, Abcam), for 1 h in a 37 °C temperature incubator. Then, the HRP was reacted with 3,3-diaminobenzidine, producing a brown coloration, which was stopped by rinsing with tap water. Finally, the sections underwent ethanol concentration dehydration, xylene clearing, and sealing with neutral resins. The images were also observed using a microscope (Nikon Eclipse, 80i, Tokyo, Japan) [42,43].

### 2.6. Western Blotting

The details of these steps have been described previously [42]. Firstly, the separated fresh tissues were cut down and put into the cold RIPA lysis buffer on ice. The ratio of tissue sample weight and buffer volume is 1 mg: 3 μL in the Eppendorf tube. This mixture was smashed by an ultrasonic crusher (E0386, Beyotime, Nanjing, China) on ice. To measure the protein concentration, protein standard solution and protein mixture (10–20 mL) were, respectively, added with 5 × CBB (Coomassie brilliant blue, 200 mL) on a 96-hole board with BCA assay (P0009, Beyotime, China). Then, the extracted protein (30 µg) was mixed with 5 × loading buffer in a centrifuge tube, boiled for 5 min to degenerate, then centrifuged (13,000 rmp, 4 °C) for 15 min. Loading samples and markers were added into the comb hole and electrophoresed on 5% concentrated SDS–polyacrylamide gels (80 V, 20 min) and 10–15% gels (120 V, 100 min). The separated proteins on the gels were transferred onto PVDF membranes at 300 mA/70 V for 2 h. The blots were sealed with a 5% blocking buffer for 1 h. The primary antibodies were added into the incubator box at 4 ◦C overnight. And the primary antibodies in this study were followed: ZO-1 (1:300, sc-33725, Santa Cruz), Occludin (1:200, sc-133256, Santa Cruz), IL-1β (1:200, sc-32294, Santa Cruz), p47^phox^ (1:300, sc-17844, Santa Cruz), and gp91^phox^ (1:300, sc-130543, Santa Cruz) in Western blotting were employed. β-actin (M00521, Genscript, Piscateville, NJ, USA) served as an internal control. After being washed with PBST, the blots were incubated for 1 h with an HRP-conjugated secondary antibody (dilution: 1:2000, Santa Cruz Biotechnology). The bands were washed with water five times and soaked with chemiluminescence (Amersham ECL Plus, Cytiva, Marlborough, MA, USA) for 2–3 min. Finally, the band was visualized and analyzed using the Chemical optometry imaging analysis system (ChemiDoc^TM^ XRS+, Bio-Rad, Hercules, CA, USA) and Image J software (NIH image program, version 1.49).

### 2.7. 16S rRNA Gene Sequencing and Gut Microbiota Analysis

Fecal samples were collected and stored at −80 °C. Then, they were packed with dry ice and shipped to GENEWIZ Biotechnology (Suzhou, China) for 16S rDNA Amplicon Sequencing analysis. Firstly, the fecal sample DNA was extracted using an Assay Kit (MP Biomedicals, Santa Ana, CA, USA), and the concentration was detected using an ultraviolet spectrophotometer. After the DNA Sequence Amplification, 1% SDS-gel electrophoresis and Nanodrop 2000 were used for Genomic DNA Quality Detection. Regarding the amplification of Target Region, the V3-V4 region of the 16S rRNA gene was amplified using Illumina primers: 5′TCGTCGGCAGCGTCAGATGTGTATAAGAGACAGCCTACGGGNGGCWGCAG and 5′GT CTCGTGGGCTCGGAGATGTGTATAAGAGACAGGGACTA CHVGGGTWTCTAAT for index PCR. Meanwhile, those DNA were added to sample-specific index sequences for Library Quantification and Pooling. Afterward, a small fragment library was detected through the NovaSeq 600 platform and SP-Xp(PE250) paired-end sequencing. The raw data were subject to quality filtering, noise reduction, splicing, and de-chimerization, which generated ASV/OUT cluster analysis by the DADA2 plug-in within the QIIME2 software (versions 2023.02). Finally, alpha diversity (Alpha Diversity) and beta diversity analysis (Beta Diversity) in the gut microbiota were analyzed according to our previous studies [17,19].

### 2.8. Untargeted Metabolomics in Faeces

The same fecal samples were weighed (20 mg), and 70% methyl alcohol (400 μL) was added to the tube. Then, the mixture was smashed by ultrasonic crusher (E0386, Beyotime, China) on ice and centrifuged (12,000 r/min, 10 min), and the supernatant (300 μL) was transferred to the new tube. This step was repeated two times to obtain the metabolite. A QTOF/MS-6545 mass spectrometer and a 1290 Infinity LC high-performance liquid chromatograph (both Agilent, Santa Clara, CA, USA) were used to detect the metabolites in the fecal samples following taurine treatment [45,46]. For liquid chromatographic condition (HILIC) separation, the samples were analyzed using the Waters ACQUITY UPLC HSS T3 C18 1.8 µm with dimensions of 2.1 mm × 100 mm (Waters, Ireland). Mobile phase A consisted of ultra-pure water (0.1% formic acid). Mobile phase B consisted of acetonitrile (0.1% formic acid). The column temperature was 40 ^∘^C; the flow rate was 0.40 mL/min (sample size: 2 μL). Under the positive ion mode mass spectrometry, the raw data were converted and filtered. The corrected and filtered peaks were identified by searching in the database and using the metDNA method.

### 2.9. ELISA

We used ELISA kits to quantify the levels of NF-κB p65 (ab133112, Abcam, Cambridge, UK), IL-1β (S0131, Abcam), TNF-α (S0073, Abcam), IL-10 (ab214566, Abcam), L-kynurenine (BA-E-2200, Immusmol, Bordeaux, France), L-tryptophan (BA-E-2700, Immusmol, France), serotonin (5-HT, abx156199, Abbexa, Cambridge, UK), and 5-hydroxyindoleacetic acid (5-HIAA, abx150308, Abbexa) levels were quantified using corresponding ELISA kits according to the manufacturer’s protocols [39].

### 2.10. Statistical Analysis

Data were presented as mean ± standard error of the mean (SEM). Statistical analyses were performed using GraphPad Prism software version 8.0. Systolic blood pressure (SBP) measurements were analyzed using repeated-measures ANOVA. One-way ANOVA followed by Tukey’s post hoc test was employed to determine significant differences in the number of positive neurons, fluorescent intensities, Western blotting data, and ELISA results. A *p*-value < 0.05 was determined as the statistical significance.

## 3. Results

### 3.1. Effects of Taurine on Blood Pressure and Cardiac Hypertrophy in Hypertensive Rats

Following taurine administration, the systolic blood pressure (SBP) in the SHR rats began to decrease at four weeks, as compared to the SHR+Taurine group (143.51 ± 1.32 mmHg vs. 182.64 ± 1.58 mmHg, *p* < 0.001). This reduction in SBP was more pronounced at eight weeks (185.72 ± 1.51 mmHg vs. 143.00 ± 2.23 mmHg, *p* < 0.001, Figure 1a). Additionally, the MAP in SHR rats was significantly higher than that in the control group (163.83 ± 1.62 mmHg vs. 93.83 ± 0.60 mmHg, *p* < 0.001, Figure 1b). Long-term taurine treatment decreased the MAP in hypertensive rats from 163.83 ± 1.62 mmHg to 140.20 ± 2.46 mmHg (*p* < 0.001). The plasma norepinephrine (NE) levels, which are a marker of sympathetic nerve activity, were also assessed. After eight weeks of taurine treatment, the plasma NE levels in the SHR+Taurine group were lower than those in the SHR group (*p* < 0.001, Figure 1c), indicating that taurine supplementation reduced both blood pressure and sympathetic nerve activity.

Additionally, we assessed the hypertension-induced cardiac hypertrophy and collagen deposition in the heart tissue using H&E and Masson’s trichrome staining so as to evaluate taurine’s effects on cardiac remodeling. Compared to the WKY group, hypertension significantly increased the cardiomyocyte cross-sectional area and perivascular fibrosis (Figure 1d). In the SHR group, taurine treatment significantly reduced the cardiomyocyte cross-sectional area (*p* < 0.05, Figure 1e) and perivascular fibrosis (*p* < 0.001, Figure 1f).

### 3.2. Effects of Taurine on Gut Microbial Composition in Hypertensive Rats

Next, the 16S rRNA gene sequencing of the fecal samples was performed for each group. The PCoA, visualized in three dimensions (*p* < 0.05, Figure 2a), revealed the distinct clustering of the intestinal microbiota among the three groups. The WKY and SHR groups exhibited no overlap, and the taurine treatment induced alterations in the intestinal microbiota of the SHR group. Moreover, the Chao1 richness (*p* < 0.05, Figure 2b) and Shannon diversity (*p* < 0.05, Figure 2c) were significantly lower in the SHR group compared to the WKY group. However, the taurine treatment significantly increased both indices in the SHR group, suggesting a potential restorative effect. To identify the specific bacterial taxa associated with the group differences, a LEfSe analysis was conducted. As shown in Figure 2d,e, long-term hypertension elevated the relative abundance of *Lactobacillaceae*, *Lactobacillales*, and *Bacilli* while increasing the relative abundance of *Oscillospiraceae*, *Lachnospiraceae*, *Lachnospirales*, *Oscillospirales*, *Ruminococcaceae*, and *Clostridia*, compared to the WKY group. Taurine supplementation restored the intestinal microbiota’s composition by increasing the relative abundance of *Lactobacillaceae*, *Lactobacillales*, and *Bacilli*; these bacterial families are known to be associated with SCFA production, particularly butyrate see Figure 2f (*p* < 0.05). The ANOSIM results also showed large differences in both the WKY vs. SHR groups and the SHR vs. SHR+Taurine groups (Table 1).

### 3.3. Effects of Taurine on Pathological Features in Hypertensive Rat Gut

Previous studies have demonstrated that long-term hypertension impacts colonic collagen deposition, the muscle layer thickness, the goblet cell number, and the crypt depth. The Masson results revealed increased collagen deposition (Fibrotic area %, *p* = 0.0006, Figure 3a,b) in the colons of SHR rats compared to the control group, and this was ameliorated by taurine treatment (*p* = 0.0323, Figure 3a,b). The H&E results showed the significant thickening of the muscle layer in the colons of SHR rats compared to WKY rats (*p* < 0.05, Figure 3c,d). Additionally, a decrease in the goblet cell area percentage (*p* < 0.05, Figure 3e,f) and crypt depth (*p* < 0.05, Figure 3g,h) was observed in the SHR group compared to the WKY group. Long-term taurine treatment reversed these alterations in the SHR rats (*p* = 0.0022, Figure 3f and *p* = 0.043, Figure 3h).

### 3.4. Effects of Taurine on ZO-1 and Occludin in Hypertensive Rat Colon

To assess the impact of taurine treatment on intestinal permeability, we used immunohistochemical staining and Western blotting. Compared to the WKY controls, the SHR group exhibited decreased intestinal tight junction protein expression, as evidenced by the reduced ZO-1 and Occludin staining (tight junction proteins) (Figure 4a). Eight weeks of taurine treatment significantly increased the ZO-1 and Occludin immunopositivity in the colon tissue of SHR rats (*p* < 0.05, Figure 4b and *p* = 0.0034, Figure 4c). The Western blotting results aligned with those of the immunofluorescence staining, indicating that taurine enhanced the colon barrier function (*p* < 0.05, Figure 4d–g).

### 3.5. Effects of Taurine on Neuroinflammation in the PVN of Hypertensive Rat

The accumulation of ROS and neuroinflammation within the nucleus paraventricularis contribute to sympathoexcitation and metabolic disorder in the pathophysiology of hypertension. The immunohistochemistry results revealed that the number of IL-1β- and TNF-α-positive staining areas in the PVN in the SHR group was significantly higher than that in the WKY group. Taurine supplementation reduced the abundance of IL-1β- and TNF-α-positive neurons compared to the hypertensive control (*p* < 0.05, Figure 5a–c). The Western blotting results demonstrated that the protein expression of the pro-inflammatory cytokines IL-1β and TNF-α was elevated in the SHR group, while the expression of the anti-inflammatory cytokine IL-10 was lower compared to the control group (*p* < 0.05, Figure 5d–g). Taurine treatment decreased the IL-1β and TNF-α protein expression and increased the IL-10 protein expression in the SHR group, which was consistent with the immunohistochemical findings.

### 3.6. Effects of Taurine on Oxidative Stress in the PVN of Hypertensive Rats

We measured the p47^phox^ expression in the PVN using immunohistochemistry and the superoxide anion levels via DHE staining across the three groups to show the taurine influence on oxidative stress. Compared to the WKY group, hypertension elevated the PVN expression of p47^phox^ and superoxide anion in SHR rats. Taurine supplementation significantly reduced the PVN expression of p47^phox^ and superoxide anion in SHR rats (*p* < 0.05, Figure 6a–c). Furthermore, the Western blotting analyses revealed increased gp91^phox^ protein expression (*p* < 0.05, Figure 6d–f) in SHR rats compared to the controls. Taurine administration decreased the gp91^phox^ protein expression in SHR rats, indicating a reduction in ROS overproduction in the PVN (*p* < 0.05, Figure 6b,c,f).

### 3.7. Effects of Taurine on Inflammation in the PVN and Colon

Using an ELISA, we measured inflammatory cytokine indicators in the PVN and colon. The ELISA results showed that the NF-κB activity, IL-1β level, and TNF-α level in the PVN and colon were significantly higher in hypertensive rats compared to the control group (*p* < 0.05, Figure 7a–d). Taurine administration significantly reduced the NF-κB activity, IL-1β level, and TNF-α level in the SHR group. Conversely, the levels of IL-10, an anti-inflammatory cytokine, were increased in the PVN and colon following taurine administration, as compared to the SHR group (*p* < 0.05, Figure 7e–h). These findings reveal that taurine supplements suppress the inflammatory responses in both the PVN and colon in hypertensive rats.

### 3.8. Multivariate Analysis of Metabolic Profiles in Gut Microbiota

Metabolomics is a quantitative analytical approach that is used to profile all of the metabolites within an organism and elucidate their relationships with physiological and pathological processes. Therefore, this study used non-targeted metabolomics to investigate the effects of taurine on the gut metabolic profiles in spontaneously hypertensive rats. The 3D PCA score plots (Figure 8a) showed a clear separation between the taurine-treated and control groups, indicating significant alterations in the levels of endogenous gut microbiota metabolites following eight weeks of oral taurine administration.

A partial least-squares discriminant analysis (PLS-DA) model was conducted as a supervised statistical method to enhance the classification accuracy and identify differential metabolites between the SHR and SHR+Taurine groups. As illustrated in Figure 8b, the score plots generated from the OPLS-DA models indicated that the samples from different groups were separated, with no overlap, demonstrating that taurine effectively altered the physiological and metabolic conditions in the SHRs.

The OPLS-DA model parameters R^2^Y and Q^2^ were calculated using cross-validation to assess the model’s interpretability and predictive ability. High R^2^Y and Q^2^ values indicate an excellent OPLS-DA model. For the SHR versus SHR+Taurine groups in positive ion mode, the R^2^Y and Q^2^ values were 0.99 and 0.869, respectively. All *p*-values for R^2^Y and Q^2^ were less than 0.005 (Figure 8c) in positive ion mode, confirming the robustness and reliability of the OPLS-DA models for biomarker and metabolic pathway analysis. Above all, long-term taurine supplementation has an effect on the metabolic states of hypertensive rats. The OPLS-DA model is highly reliable, making it suitable for the further analysis of biomarkers and metabolic pathways.

### 3.9. Identification of Potential Biomarkers and Metabolic Pathways

A total of 686 potential biomarkers were identified in the positive ion model comparing the SHR and SHR+Taurine groups, with 561 up-regulated and 125 downregulated biomarkers (Table 2). Volcano plots were generated to visualize the metabolic differences between the positive and negative ion models (Figure 8d). In such plots, points located farther from the center indicate a more significant contribution to the metabolic differences than those closer to the center. Metabolites located nearer to the plot’s corners are considered more important.

To identify potential biomarkers and elucidate the underlying metabolic pathways, KEGG enrichment analysis was used to analyze the change in metabolites. The image (Figure 8e) indicated that tryptophan metabolism was the primary metabolic pathway influenced by taurine intervention in the positive ion model.

### 3.10. Effects of Taurine on Tryptophan Metabolism in Hypertensive Rat Serum

To further investigate tryptophan metabolism, we assessed tryptophan-related markers. The results demonstrated that hypertension decreased the serum kynurenine levels and the Kyn/Try ratio (*p* < 0.05, Figure 9a,c) while increasing the serum L-tryptophan, 5-HT, and 5-HIAA levels (*p* < 0.05, Figure 9b,d,e). Taurine treatment elevated the serum kynurenine levels and Kyn/Try ratios and reduced the serum L-tryptophan, 5-HT, and 5-HIAA levels compared to the hypertensive group.

## 4. Discussion

In this study, we used a spontaneously hypertensive rat, which not only exhibits the characteristics of spontaneous hypertension but also displays a similar pathogenesis to human hypertension. According to the results, long-term hypertension-induced cardiac damage, activated inflammatory responses, ROS overproduction in the PVN, and disrupted gut microbiota equilibrium all contribute significantly to the hypertensive process. It is notable that taurine supplementation not only reduced the animals’ blood pressure and sympathetic activity but also ameliorated the cardiac pathology, restored the microbiota’s composition, enhanced intestinal homeostasis and barrier integrity, and mitigated gut inflammation, collectively preventing the pathological changes caused by hypertension. Furthermore, the taurine treatment suppressed inflammation and the overproduction of ROS in the PVN in SHRs. The untargeted metabolite analysis revealed that tryptophan metabolism and the associated metabolic profiles were the most likely targets of the taurine intervention in SHRs. Therefore, taurine supplementation restores the microbiota balance, strengthens the mucosal barrier, reduces intestinal inflammation, and stimulates tryptophan metabolism. Moreover, the metabolites derived from the gut microbiota traverse the blood–brain barrier to reach the paraventricular nucleus, thereby reducing the inflammatory responses and oxidative stress, leading to decreased blood pressure in hypertensive rats.

Firstly, this study confirms that taurine has positive effects in reducing arterial pressure and exhibits potential as an anti-hypertensive agent, which is consistent with previous findings [47,48,49]. Zhu et al. (2016) also reported that taurine decreased blood pressure and improved vascular function in the context of prehypertension, potentially through reduced agonist-induced vascular reactivity [50]. In addition, the intracellular cytokine levels and ROS overload in the endocardial endothelium were reduced with taurine supplementation in hypertensive rats [51], contributing to its overall beneficial effects on cardiomyocyte health. Therefore, taurine administration significantly reduces the sympathetic activity in spontaneously hypertensive rats, concurrently ameliorating cardiomyocyte hypertrophy and fibrosis during the development of hypertension.

A growing body of research links hypertension to alterations in the intestinal flora distribution [15,17,43]. Using 16S rRNA gene sequencing, we observed that chronic hypertension disrupted the richness and diversity of the microbiota compared to normotensive controls (WKY). While WKY rats exhibited a predominance of *Clostridia*, Lactobacillaceae, and Oscillospiraceae, SHRs displayed increased abundances of Bacilli, *Lactobacillaceae*, *Lactobacillales*, *Prevotella*, and *Staphylococcaceae*. Notably, the levels of *Clostridia* and *Oscillospiraceae* were reduced in the SHR group. This demonstrates that morbid intestinal flora influenced the blood pressure fluctuation in the host [16]. Iñaki Robles-Vera (2017) reported that fecal microbiota transplantation from adult SHRs to adult WKY rats induced a sustained increase in systolic pressure. Conversely, fecal microbiota transplantation from normal rats to adult hypertensive rats decreased blood pressure and improved endothelial function [52]. There are bidirectional effects on the fecal microbiota composition and blood pressure regulation. Chronic hypertension impaired the colonic structure, including excessive collagen deposition and muscle layer thinning, along with reduced and shortened goblet cells, indicating colonic damage. The mucosal barrier, composed of intact intestinal epithelial cells and tight junctions, is crucial for intestinal integrity. In our study, hypertension weakened tight junction proteins such as ZO-1 and Occludin, which led to increased intestinal permeability and bacterial translocation. These components leak into the blood circulation due to the increase in intestinal permeability, worsening the systemic inflammation [53,54]. However, treatment with taurine ameliorated the tight junction protein expression and reduced the colonic inflammation in SHRs. Levy et al. (2015) demonstrated that taurine, as a host microbiota metabolite, suppressed inflammation and maintained the microbial community’s stability, preventing dysbiosis-related diseases [55]. This is consistent with our findings. Meanwhile, in line with the gut microbiota results, taurine supplementation increased the abundance of *Oscillospiraceae*, *Lactobacillales*, *Clostridia*, and *Ruminococcaceae* in the SHRs; these are known to produce SCFAs, with beneficial effects on intestinal epithelial cell growth, barrier function, and anti-inflammation, probably through the transport of MCT1 and MCT4 and connection with the receptors GPR41 and GPR43 [16]. Our findings also indicate that taurine supplementation restored the intestinal flora and corrected the microbiota imbalance. These effects likely attenuated the sympathetic nerve outflow and regulated the arterial tone.

Numerous studies performed by our lab have demonstrated that chronic hypertension induces inflammatory responses and ROS overproduction in the PVN, leading to increased hypertensive reaction. The recent document confirms that the central transmits the information to gut microbiota through the autonomic nervous system [56]. With hypertension, gut dysbiosis can evoke inflammation, including host systemic inflammation and/or neuroinflammation [57]. In the present study, taurine supplementation decreased the PVN levels of IL-1β and TNF-α, the expression of NAD(P)H subunits (p47^phox^ and gp91^phox^), and the levels of ROS in spontaneously hypertensive rats. To estimate the taurine’s anti-hypertensive mechanisms in the gut microbiota and its ability to compensate for cardiovascular system damage, we employed untargeted metabolomics. The KEGG enrichment analysis revealed tryptophan metabolism as the primary metabolic pathway affected by the taurine intervention. This pathway encompasses the indole, serotonin, and kynurenine branches. We assessed the serum levels of kynurenine, Kyn/Try, L-tryptophan, 5-HT, and 5-HIAA in the different groups and found elevated levels of markers associated with all three pathways in the taurine-treated SHRs compared to the untreated SHRs. Zhang et al. (2023) reported that the microbiota-derived tryptophan metabolite indole-3-lactic acid ameliorated intestinal ischemia/reperfusion injury by repairing epithelial cell damage, reducing oxidative stress, and exerting anti-inflammatory effects [58]. Indole-3-acetate (I3A) has been shown to significantly reduce TNF-α, IL-1β, and MCP-1 expression in a dose-dependent manner [59]. Fermentation studies have demonstrated that Clostridium produces 3-indoleacrylic acid (IA) and indole-3-propionic acid (IPA) [60,61], aligning with our 16S rRNA results showing the decreased abundance of Clostridia in SHRs and its increase following taurine treatment. Regarding the kynurenine pathway, Lionetto et al. (2021) emphasized the role of altered kynurenine metabolism in inducing immune reactivity and triggering inflammation [62]. In our study, taurine administration reduced the serum kynurenine levels and the kynurenine/tryptophan ratio in SHRs. Meanwhile, 5-HT acting with gut vagal afferents can regulate the enteric nervous system, gut movement, and inflammatory reaction [63]. Stasi et al. (2019) highlighted the pleiotropic role of 5-HT in gastrointestinal and neurological/psychiatric functions [64]. They also suggested a link between serotonin metabolism and the transition of the gut microbiota from a homeostatic to an inflammatory state. Numerous studies have shown that chemical and mechanical sensors under the lamina propria and crypts of the gastrointestinal tract pass the signal to the CNS. Therefore, taurine ameliorated tight junction protein expression, reduced colonic inflammation, and improved the intestinal microenvironment via the autonomic nervous system in response to the CNS so as to suppress the inflammation and oxidative stress in the PVN. It is also possible that part of the taurine-induced tryptophan enters the bloodstream and arrives at the brain, decreasing the levels of inflammatory cytokines and ROS in the PVN. Another group of metabolites transmits afferent sensory information via anti-hypertensive effects on the CNS through the enteric nervous system (ENS) and the vagal nerve.

In conclusion, taurine supplementation effectively restored the microbiota balance; reinforced the mucosal barrier; mitigated intestinal inflammation; and stimulated tryptophan metabolism through the indole, serotonin, and kynurenine pathways. The resulting metabolites, possessing antioxidant and anti-inflammatory properties, were released into the serum or enabled the transmission of afferent sensor information to the CNS, where they suppressed inflammatory responses and ROS overproduction in the PVN. Ultimately, this results in sympathetic hypertarachia, reduced blood pressure, and attenuated cardiovascular damage. Overall, these findings enable a novel understanding of gut–brain communication in the context of hypertension under taurine treatment. Moreover, they shed new light on taurine’s anti-hypertensive effects and provide innovative ideas for the clinical treatment of hypertension.

## 5. Limitation

This study had some limitations. One concern is that cardiovascular function should not be only measured in terms of systolic pressure and MAP. Other cardiac parameters, including the left ventricular systolic pressure (LVSP), left ventricular diastolic pressure (LVDP), left ventricular end-diastolic pressure (LVEDP), maximum rise or drop rate of left ventricular pressure (dp/dtmax, -dp/dtmax), should also be analyzed. This could lead to a more comprehensive understanding of the effects of taurine on cardiovascular function during hypertension. In addition, this study was limited in establishing the gut–brain axis. We only focused on the changes in tryptophan in the feces and blood. However, it is still necessary to explore how the metabolites from the gut cross the blood–brain barrier and reach the brain, act upon the ROS and inflammation in the PVN, and then suppress the hypertensive responses. This would help to further examine the gut–brain communication during the development of hypertension. In the future, additional work will be necessary. Regarding metabolomics, the targeted metabolism of tryptophan should be further explored as the tryptophan pathway plays a crucial role in the communication within the gut–brain axis. Meanwhile, tryptophan supplementation can be applied in SHRs to verify the effect of tryptophan in terms of reducing blood pressure.

## Figures and Tables

**Figure 1 biomedicines-12-02711-f001:**
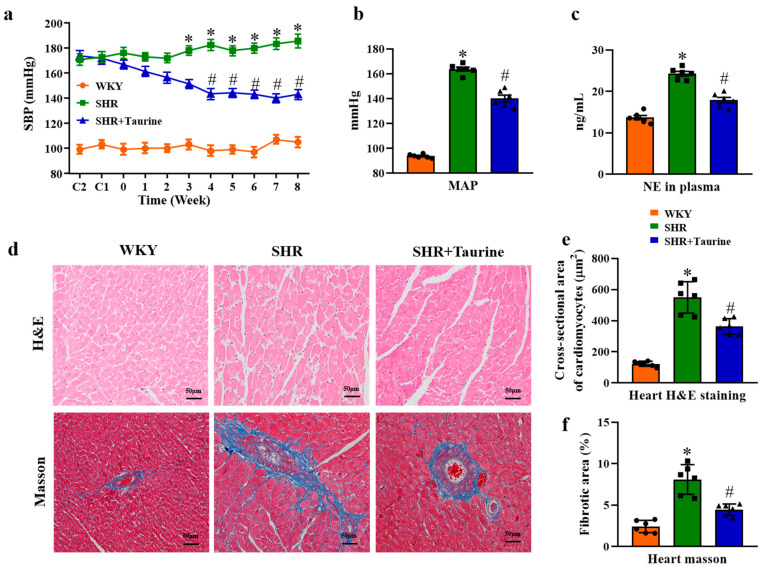
Effects of taurine on blood pressure and cardiac hypertrophy in SHR and WKY rats: (**a**) Systolic BP (SBP) in 8 weeks; (**b**) mean arterial blood pressure (MAP); (**c**) noradrenaline (NE) in plasma; (**d**) the images of H&E and Masson’s trichrome staining for heart; (**e**) histogram of cross-sectional area of cardiomyocytes; (**f**) histogram of perivascular fibrosis. ● represents WKY group data, ■ represents SHR group data, ▲ represents SHR+Taurine group data. Values are expressed as the mean ± SEM. One-way ANOVA with Tukey’s multiple comparison tests (n = 6). * *p* < 0.05 versus WKY groups; *^#^ p* < 0.05 SHR vs. SHR+Taurine.

**Figure 2 biomedicines-12-02711-f002:**
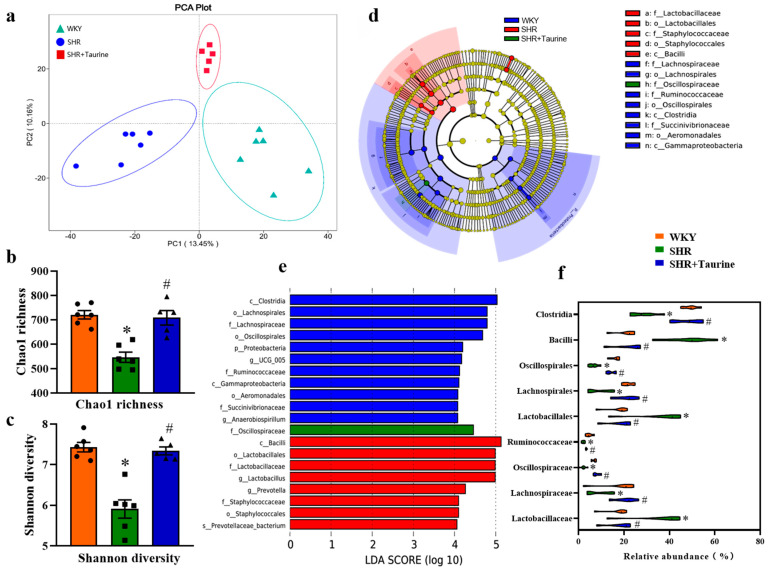
Effects of taurine on gut microbial composition in hypertensive rats: (**a**) Principal Co-ordinates Analysis (PCoA) of microbiota, PCA plots of the first two principal components for three groups. Ellipses around points represent a 95% confidence interval. (**b**) Chao 1 richness; (**c**) Shannon diversity; (**d**) linear discriminant analysis effect size (LefSe) analysis, and each circle from the inside to out represents Phylum, Class, Order, Family, Genus, Species, but the yellow dots represent those species have no significant with others; (**e**) the cladograms of intestinal flora; (**f**) relative abundance of *Clostridia*, *Bacilli*, *Oscillospirales*, *Lachnospirales*, *Lactobacillales*, *Ruminococcaceae*, *Oscillospiraceae, Lachnospiraceae*, *Lactobacillaceae*, and the more data distribution, the bigger the violin area. ● represents WKY group data, ■ represents SHR group data, ▲ represents SHR+Taurine group data. Values are expressed as the mean ± SEM. One-way ANOVA with Tukey’s multiple comparison tests (n = 6 in WKY and SHR group. n = 5 in SHR+Taurine group). * *p* < 0.05 versus WKY groups; *^#^ p* < 0.05 SHR vs. SHR+Taurine.

**Figure 3 biomedicines-12-02711-f003:**
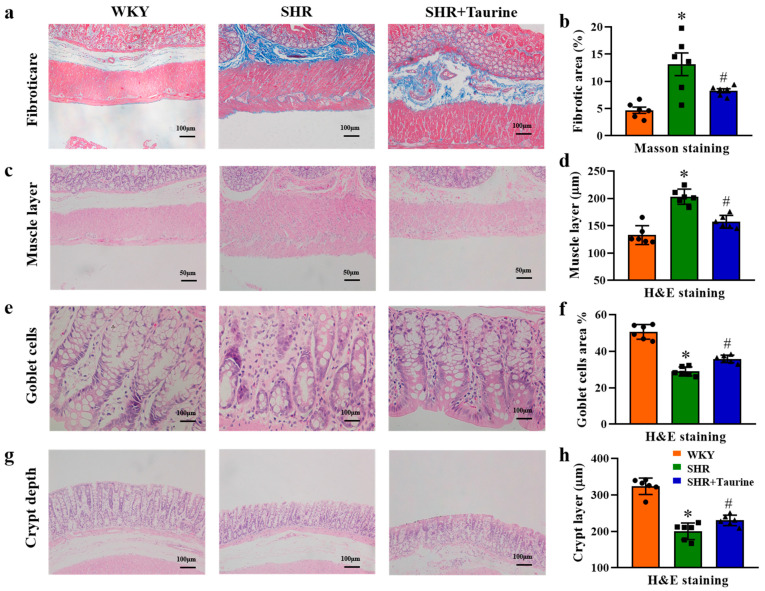
Histopathological analysis of the effects of taurine in hypertensive rat colon: (**a**) Masson’s trichrome staining for collagen deposition and (**b**) histogram of Fibroticare area %; (**c**) H&E staining for the muscles layer thickness and (**d**) analysis of muscle layer thickness; (**e**) H&E staining for goblet cells and (**f**) histogram of goblet cells area %; (**g**) H&E staining for crypt depth and (**h**) analysis of crypt depth. ● represents WKY group data, ■ represents SHR group data, ▲ represents SHR+Taurine group data. Values are expressed as the mean ± SEM. One-way ANOVA with Tukey’s multiple comparison tests (n = 6). * *p* < 0.05 versus WKY groups; *^#^ p* < 0.05 SHR vs. SHR+Taurine.

**Figure 4 biomedicines-12-02711-f004:**
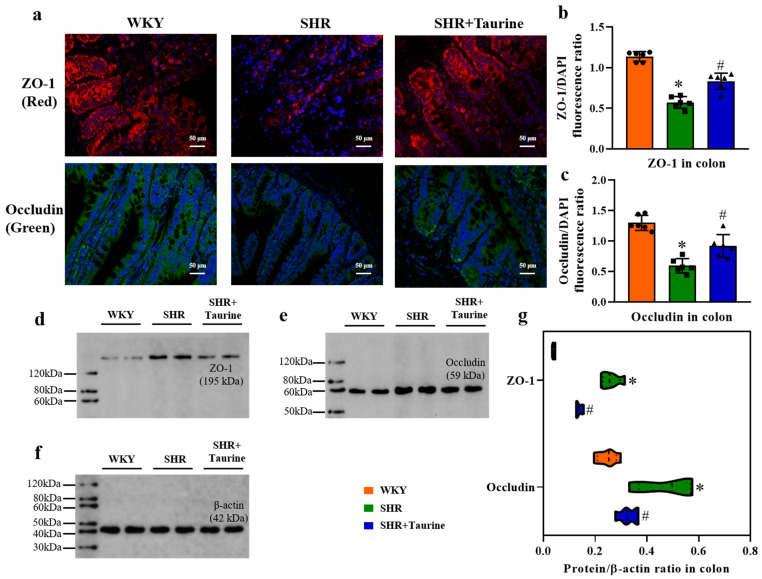
Effects of taurine on ZO-1 and Occludin in hypertensive rat colon: (**a**) Immunohistochemical staining for ZO-1 (red) and Occludin (green) in the colon, DAPI is blue; (**b**) histogram of ZO-1/DAPI fluorescence ratio and (**c**) Occludin fluorescence ratio; Western blotting for ZO-1 and (**d**–**f**) Occludin; (**g**) Violin of protein/β-actin ratio (n = 5), and the more data distribution, the bigger the violin area. ● represents WKY group data, ■ represents SHR group data, ▲ represents SHR+Taurine group data. Values are expressed as the mean ± SEM. One-way ANOVA with Tukey’s multiple comparison tests (n = 5/6). * *p* < 0.05 versus WKY groups; *^#^ p* < 0.05 SHR vs. SHR+Taurine.

**Figure 5 biomedicines-12-02711-f005:**
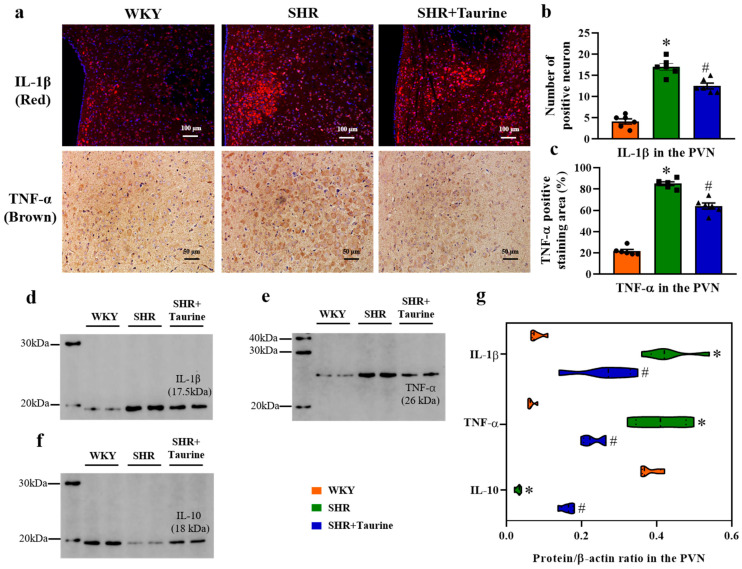
Effects of taurine on neuroinflammation in the PVN of a hypertensive rat: (**a**) immunofluorescence for IL-1β (red), DAPI is blue and immunohistochemical staining TNF-α (brown) in the PVN; (**b**) IL-1β in the PVN histogram of the number of positive neurons and (**c**) TNF-α in the PVN histogram of positive staining area %; (**d**–**f**) Western blotting for IL-1β, TNF-α, and IL-10 in the PVN; (**g**) Violin of protein/β-actin ratio (n = 5). ● represents WKY group data, ■ represents SHR group data, ▲ represents SHR+Taurine group data. Values are expressed as the mean ± SEM. One-way ANOVA with Tukey’s multiple comparison tests (n = 5/6). * *p* < 0.05 versus WKY groups; *^#^ p* < 0.05 SHR vs. SHR+Taurine.

**Figure 6 biomedicines-12-02711-f006:**
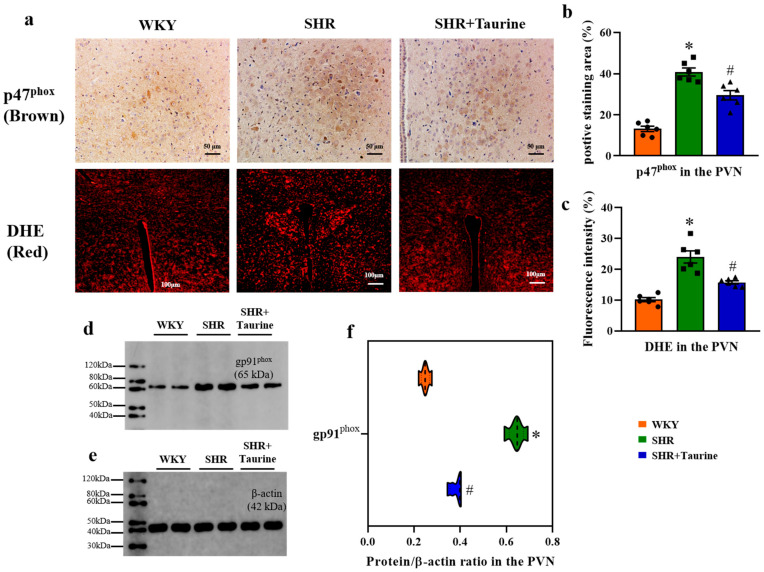
Effects of taurine on oxidative stress in the PVN of hypertensive rat: (**a**) immunohistochemical staining p47^phox^ (brown) in the PVN and superoxide anion staining (red) DHE; (**b**) histogram of positive staining area % and (**c**) fluorescence intensity %; (**d**,**e**) Western blotting for gp91^phox^; (**f**) Violin of protein/β-actin ratio (n = 5), and the more data distribution, the bigger the violin area. ● represents WKY group data, ■ represents SHR group data, ▲ represents SHR+Taurine group data. Values are expressed as the mean ± SEM. One-way ANOVA with Tukey’s multiple comparison tests (n = 5/6). * *p* < 0.05 versus WKY groups; *^#^ p* < 0.05 SHR vs. SHR+Taurine.

**Figure 7 biomedicines-12-02711-f007:**
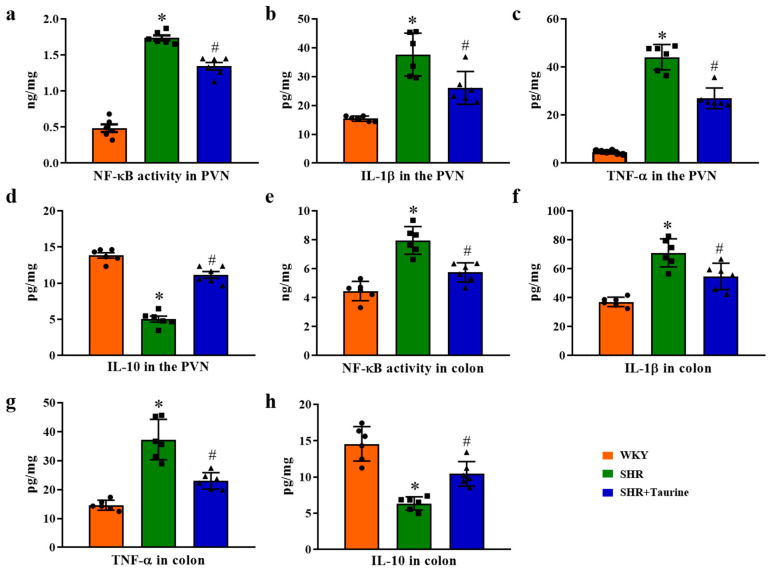
Effects of taurine on inflammation in the PVN and colon: ELISA analysis of inflammation in the PVN and colon. Histogram of (**a**) NF-κB activity, (**b**) IL-1β level, (**c**) TNF-α level, and (**d**) IL-10 level in the PVN. Histogram of (**e**) NF-κB activity, (**f**) IL-1β level, (**g**) TNF-α level, and (**h**) IL-10 level in colon. ● represents WKY group data, ■ represents SHR group data, ▲ represents SHR+Taurine group data. Values are expressed as the mean ± SEM. One-way ANOVA with Tukey’s multiple comparison tests (n = 6). * *p* < 0.05 versus WKY groups; ^#^
*p* < 0.05 SHR vs. SHR+Taurine.

**Figure 8 biomedicines-12-02711-f008:**
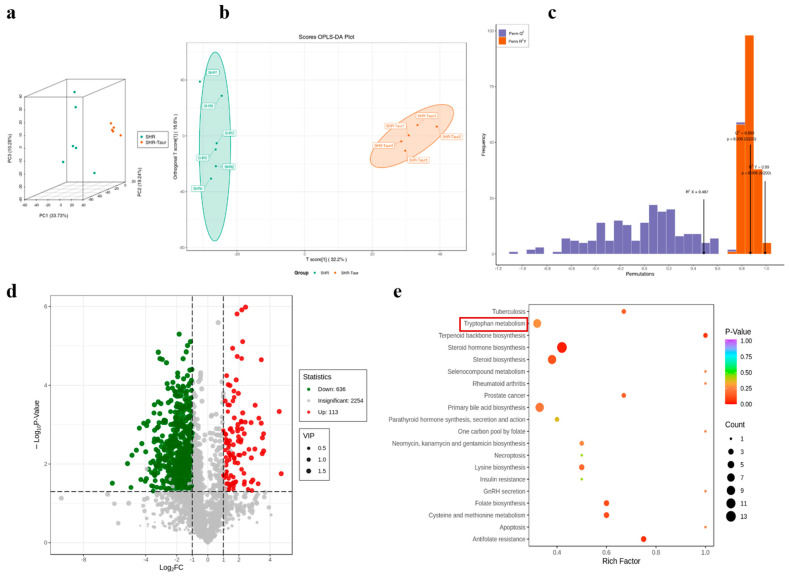
Multivariate analysis of metabolic profiles in gut microbiota: (**a**) PCA-3D. (**b**) Scores OPLS-DA Plot: X—first principal component; Y—the second principal component. Ellipses around points represent a 95% confidence interval. (**c**) OPLS-Permutation, Q^2^ > 0.9 means a reliable model, blue represents the Perm Q^2^, orange represents the Perm R^2^Y. R^2^X = 0.487; Q^2^ = 0.869 (*p* < 0.005), R^2^Y = 0.99 (*p* < 0.005). (**d**) Volcano plots: Above −Log_10_P-Value = 1.30, the dots (on the right side of Log_2_FC = 1 and on the left side of Log_2_FC = −1) are marked as significant differences metabolites between SHR group and SHR+Taurine group. The dot is greater than 2 times (on the right side of Log_2_FC= 1) as red, and less than −2 (on the left side of Log_2_FC = −1) as green. Red—up-regulated metabolites; green—down-regulated metabolites. (**e**) Kyoto Encyclopedia of Genes and Genomes (KEGG) enrichment analysis of feces samples collected from the SHR group and SHR+Taurine group based on LC-MS/MS in positive ion modes. Dot size: impact value; dot color: *p*-value. (n = 6 in WKY and SHR group. n = 5 in SHR+Taurine group).

**Figure 9 biomedicines-12-02711-f009:**
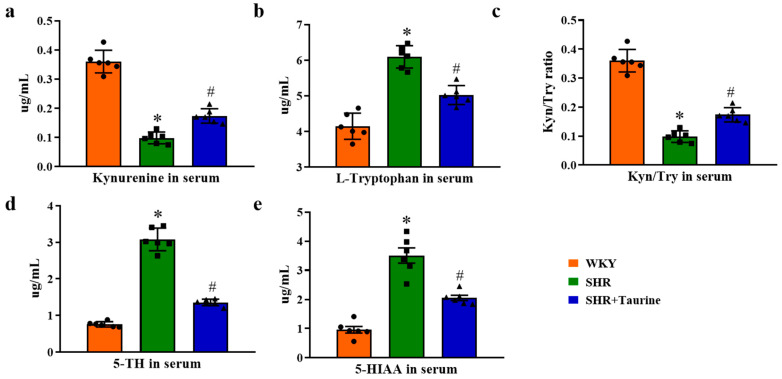
Effects of taurine on Tryptophan metabolism in hypertensive rat serum: ELISA analysis of tryptophan-related indicators in serum. Histogram of (**a**) Kynurenine, (**b**) L-Tryptophan, (**c**) Kyn/Try ratio, (**d**) serotonin (5-TH), and (**e**) 5-Hydroxyindole-3-acetic acid (5-HIAA). ● represents WKY group data, ■ represents SHR group data, ▲ represents SHR+Taurine group data. Values are expressed as the mean ± SEM. One-way ANOVA with Tukey’s multiple comparison tests (n = 6). * *p* < 0.05 versus WKY groups; *^#^ p* < 0.05 SHR vs. SHR+Taurine.

**Table 1 biomedicines-12-02711-t001:** Anosim.

Group 1	Group 2	R-Value	*p*-Value
WKY	SHR	0.8833	0.004
SHR	SHR+Taurine	0.616	0.005
SHR+Taurine	WKY	0.6907	0.004

Similarity of each of the pairing groups is shown with R- and *p*-values. R closer to 1 suggests larger differences in the two groups than the members of each group (n = 6 in WKY and SHR group. n = 5 in SHR+Taurine). *p* < 0.05 represents significance.

**Table 2 biomedicines-12-02711-t002:** Statistical table of differential metabolite numbers (positive).

Group Name	Total SigMetabolites	Down-Regulated	Up-Regulated
WKY vs. SHR	678	379	299
WKY vs. SHR+Taurine	749	636	113
SHR vs. SHR+Taurine	686	561	125

The change in metabolites between the two groups under the positive condition (n = 6 in WKY and SHR group. n = 5 in SHR+Taurine).

## Data Availability

Data of this study are available upon request to the corresponding authors.

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
