# Peer review of "Taurine Supplementation Alleviates Blood Pressure via Gut–Brain Communication in Spontaneously Hypertensive Rats"

_biomedicines, 2024, doi:10.3390/biomedicines12122711_

Round 1

Reviewer 1 Report

Comments and Suggestions for Authors

Although the authors have done extensive work, the objective of the study remains unclear. There are a number of already published works on Taurine. The authors should clarify the need and purpose of this work and how it is different from other published works.

The introduction lacks depth. The significance of Taurine concerning the work needs to be justified.

The methodology section needs to be a little more descriptive.

In Figure 1, H&E staining for the first (WKY) and third (SHR+Taurine) looks very similar. The study should be repeated to examine the results

The quality of Figure 8 should be improved

The authors should thoroughly revise the manuscript to recheck the statistical values and rectify the grammatical errors in the manuscript.

Comments on the Quality of English Language

Minor revisions are required to improve the language of the work. Additionally, some typo errors and grammatical issues need to be rectified.

Reviewer 2 Report

Comments and Suggestions for Authors

The article as a whole is interesting, and the specific details of the review are in the Word document.

Reviewer 3 Report

Comments and Suggestions for Authors

The authors of the submitted manuscript examined the effect of taurine supplementation on blood pressure in vivo. However, there are few critical points which represent the principle of the experiment and the authors have missed them. Firstly, the authors did not mention the accurate number of rats used in the study. Secondly, the accurate amount of taurin is not mentioned but only the concentration. Thirdly, while the authors are trying to investigate the gut-brain interaction on blood pressure, using SHR model to reflect human hypertension and gut-brain axis was not mentioned. Fourthly, the study also did not mention exactly the amount of water consumed by the rats which could be monitored to ensure consistent dosing among groups. 

Comments on the Quality of English Language

Should be improved 

Round 2

Reviewer 1 Report

Comments and Suggestions for Authors

Although the authors tried to address the concerns, the plagiarism of the revised version is too high. I advise the authors to carefully re revise the manuscript.

Author Response

Dear Editor and Reviewers,

We are so sorry about this accident. We had seriously re-revised the manuscript especially in “Material and methods” section and highlighted all the revision in blue colour. Please see the attachment. We appreciate your kindness and hope that the revision would meet with approval. 

Sincerely,

Dr. Qing Su and all authors

On behalf of Dr. Meng-Lu Xu

Department of Nephrology,

the First Affiliated Hospital of Xi'an Medical University,

Xi'an 710077, China.

Phone: +86 2982657490

E-mail: zhulinyanbo@163.com

Reviewer 3 Report

Comments and Suggestions for Authors

The authors respond to the comment correctly

Author Response

Dear Editor and Reviewers,

We really appreciate for your warm earnestly and reply timely. And in the future research work, we would pay much more attention to your valuable suggestions you gave us. Once again, thank you so much for your comments and suggestions in this study.

Best regards,

Dr. Qing Su and all the authors

On behalf of Dr. Meng-Lu Xu

Department of Nephrology,

the First Affiliated Hospital of Xi'an Medical University,

Xi'an 710077, China.

Phone: +86 2982657490

E-mail: zhulinyanbo@163.com